# Video-Language Critic: Transferable Reward Functions for Language-Conditioned Robotics

**Minttu Alakuijala[1]    Reginald McLean[2]    Isaac Woungang[2]    Nariman Farsad[2]**
**Samuel Kaski[1,3]    Pekka Marttinen[1]    Kai Yuan[4]**

[1]*Department of Computer Science, Aalto University*
[2]*Department of Computer Science, Toronto Metropolitan University*
[3]*Department of Computer Science, University of Manchester*
[4]*Intel Corporation*

*minttu.alakuijala@aalto.fi*

Reviewed on OpenReview: *https://openreview.net/forum?id=jJOVpnNrEp*

## Abstract

Natural language is often the easiest and most convenient modality for humans to specify tasks for robots. However, learning to ground language to behavior typically requires impractical amounts of diverse, language-annotated demonstrations collected on each target robot. In this work, we aim to separate the problem of *what* to accomplish from *how* to accomplish it, as the former can benefit from substantial amounts of external observation-only data, and only the latter depends on a specific robot embodiment. To this end, we propose Video-Language Critic, a reward model that can be trained on readily available cross-embodiment data using contrastive learning and a temporal ranking objective, and use it to score behavior traces from a separate actor. When trained on Open X-Embodiment data, our reward model enables 2x more sample-efficient policy training on Meta-World tasks than a sparse reward only, despite a significant domain gap. Using in-domain data but in a challenging task generalization setting on Meta-World, we further demonstrate more sample-efficient training than is possible with prior language-conditioned reward models that are either trained with binary classification, use static images, or do not leverage the temporal information present in video data.[1]

## 1 Introduction

Advances in natural language processing and vision-language representations have enabled a significant increase in the scalability and generalization abilities of learned control policies for robotics. Methods involving large architectures, such as Transformers (Vaswani et al., 2017), and internet-scale pretraining have transferred well to both high-level (Liang et al., 2022; Vemprala et al., 2023) and low-level (Brohan et al., 2022; Lynch et al., 2022; Shridhar et al., 2022) robotic control. Natural language has many desirable features as a modality for specifying tasks. Unlike structured, hand-designed task sets, natural language is unrestricted and open-domain. Moreover, prompts can be specified as precisely or vaguely as appropriate. While goal images, demonstration videos, or goal states more broadly, have been considered as an alternative open-domain task definition modality (Chen et al., 2021; Alakuijala et al., 2023; Ma et al., 2023b), they typically have to specify irrelevant environment details, such as the background. Furthermore, language readily supports task definitions with novel combinations of actions, objects and their attributes, as well as subtask sequencing, in a way that facilitates the policy's understanding of unseen tasks.

---

[1]Source code and supplementary videos are available on the project website: `https://sites.google.com/view/video-language-critic`.

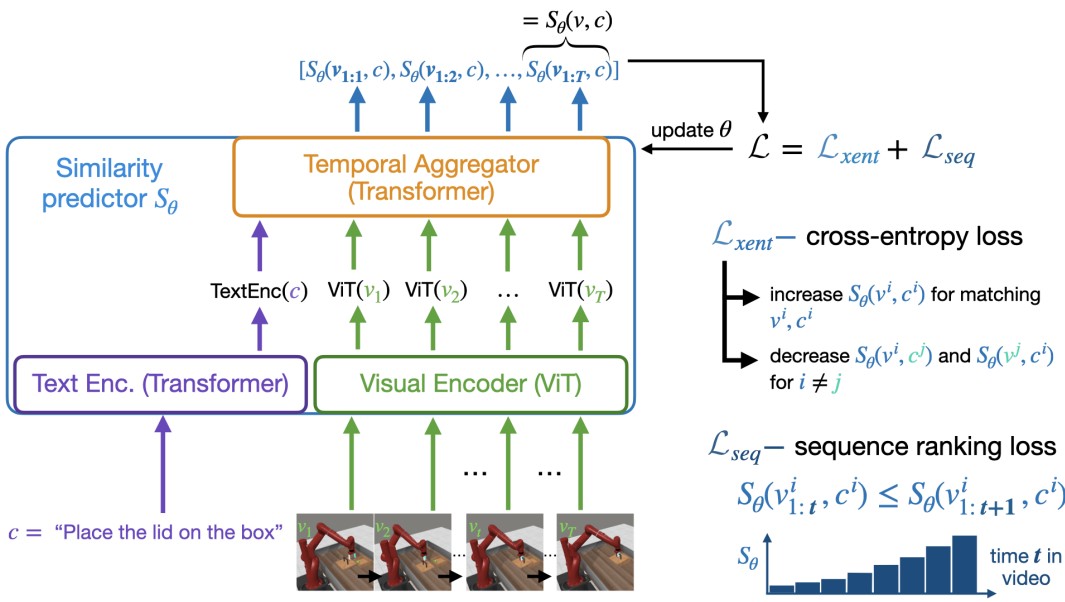

Figure 1: **Overview:** Our similarity function, $S_\theta$, is trained using video-caption pairs $(v_{1:T}^i, c^i)$. The visual encoder (ViT; Dosovitskiy et al. (2021)) is applied separately to each video frame $v_t$ to produce a sequence of image features, which are appended to the caption embedding produced by the text encoder. The temporal aggregator then predicts a similarity score for each time step $t$ of the video. We use cross-entropy and sequence ranking objectives to encourage the predicted scores to be high for matching video-caption pairs and low for mismatching pairs, and to monotonically increase over a successful execution.

Most prior work has proposed to learn language-conditioned policies end-to-end, i.e., directly predicting an action in the robot's action space given the current state and task description. However, this has several downsides: first, fitting large models on the full problem requires a significant amount of high-quality demonstration data from the target robot domain. Second, the resulting policy depends entirely on the specific robot instance, observation space, and controller type (e.g., joint or task space control) and does not easily transfer to other settings. Moreover, much of the prior work addresses vision-language grounding in robotics purely with imitation learning (Brohan et al., 2022; Lynch et al., 2022; Shridhar et al., 2022; Open X-Embodiment Collaboration et al., 2023), without attempting to discriminate between low-quality and expert demonstrations. As a result, the resulting policies are inherently limited by the skills of the demonstrators, and no novel solutions can be discovered through planning or interaction. This line of work overlooks performance gains that could be obtained by converting the language prompts to a scalar reward function. Manually defining a well-specified dense reward function to communicate task success is typically laborious and error-prone, and must be repeated for each task. To make progress towards a general-purpose robotic system that can learn human-level skills both in terms of quality (dexterity, robustness) and variety of skills, we argue these systems will need to be able to critique their own behavior, by learning reward functions at scale.

We address this problem by learning a foundation *video-language-conditioned reward model*, i.e., a critic that evaluates the progress (in the form of a video) of a task, given as a human-language instruction, and assigns a reward based on how close the robot is to completing the task. By leveraging large cross-task pretraining data, which may come from a variety of robots, our Video-Language Critic (VLC) can learn to score the alignment between a textual description and task execution regardless of the specific robot embodiment. Our experimental evaluation on Meta-World (Yu et al., 2019) manipulation tasks shows that VLC can learn useful general-purpose reward functions not only from in-domain, but also out-of-domain data (from Open X-Embodiment Collaboration et al. (2023)) collected from different robot embodiments. In Section 4, we show that VLC 1) accelerates the training of a wide range of manipulation tasks and 2) enables zero-shot learning on unseen tasks, when combined with a sparse task completion signal.

Recent work in language-conditioned rewards for robotics has used either binary classification (Shao et al., 2021; Silva et al., 2021; Nair et al., 2022a), contrastive vision-language alignment (Nair et al., 2022b; Ma et al., 2023a; Sontakke et al., 2023) or reconstruction (Karamcheti et al., 2023) objectives. However, they have not fully leveraged temporal ordering of frames to encourage increasing scores over successful episodes. In contrast, VLC provides a dense reward evaluating in-episode progress, learned through contrastive ranking. The advantages and contributions of our approach are as follows:

- We learn vision-language manipulation using actor-agnostic videos and instructions at scale without requiring tedious demonstration collection on a specific robot. Unlike end-to-end policy learning, our method can learn from cross-embodiment data without action labels.

- Through maximizing the learned reward, our policies can improve over suboptimal demonstrations, by executing the task faster or by finding better solutions.

- Our method, VLC, enables a 3x sample efficiency gain over a sparse task-completion reward, or 2x when trained exclusively on cross-embodiment data with a significant domain gap.

- VLC generalizes to unseen tasks through large-scale pretraining and language conditioning and leads to faster policy training than 5 prior reward learning methods.

- VLC is agnostic to the type of policy learning, and can be combined with model-free or model-based reinforcement learning, affordance-based grasping or model-predictive control.

## 2 Related Work

**Vision-language imitation** Many prior works have aimed to connect language instructions and vision-based observations in robotics (Lynch & Sermanet, 2020; Brohan et al., 2022; Lynch et al., 2022; Shridhar et al., 2022; Guhur et al., 2023) and in video games (Fan et al., 2022), mostly through large-scale demonstrations (Lynch & Sermanet, 2020; Brohan et al., 2022; Fan et al., 2022; Lynch et al., 2022) or pretraining (Shridhar et al., 2022; Guhur et al., 2023). However, the majority of approaches have considered imitation-based objectives only, without scoring the quality of existing trajectories or attempting to outperform prior data. We instead propose to learn a state-value function from cross-domain offline behavior, which can be optimized using either online, offline or model-based policy training.

**Multi-modal representations** Pretrained vision-language representations (Radford et al., 2021) have been adapted to a wide range of downstream tasks (Shridhar et al., 2022; Guhur et al., 2023). Shridhar et al. (2022) propose to augment pretrained CLIP (Radford et al., 2021) with Transporter nets (Zeng et al., 2021) to handle fine-grained spatial awareness required for precise manipulation. Xiao et al. (2022) train a CLIP-like contrastive embedding space from crowd-sourced language annotations for trajectories from the robot. We draw inspiration from these works, but instead define an embodiment-agnostic, language-conditioned reward function, which supports improvement over demonstration data.

**Video retrieval** Our work is related to video retrieval as we seek to move beyond image-language correspondence and match task descriptions with history-aware state representations. As learning representations across time is computationally expensive, many prior works have proposed to start from pretrained image-language representations and aggregate them over time, while fine-tuning the aggregation function's weights on video retrieval (Bain et al., 2022; Luo et al., 2022; Lu et al., 2023). Unlike in video retrieval, we aim to not only assign high alignment scores to full videos, but provide smoothly increasing reward over the whole video to indicate task progress.

**Inverse RL** Several works have proposed to infer the reward function of a task using examples of expert behavior, and to train an RL policy to optimize this reward (Russell, 1998). Most relevantly to our setting, a line of prior inverse RL methods considers the case where the observed behavior is not annotated with actions and may come from different action and observation spaces altogether, typically a human demonstrator (Sermanet et al., 2018; Schmeckpeper et al., 2020; Chen et al., 2021; Shao et al., 2021; Silva et al., 2021; Nair et al., 2022a; Zakka et al., 2022; Alakuijala et al., 2023; Ma et al., 2023b). Many of these works use either a

goal image (Zakka et al., 2022; Alakuijala et al., 2023; Ma et al., 2023b) or a demonstration video (Sermanet et al., 2018; Chen et al., 2021) rather than language conditioning, and some are only applicable for data from a single task at a time (Schmeckpeper et al., 2020; Zakka et al., 2022). Moreover, handling multi-task reward learning with an additional task identifier state variable, as done by Chen et al. (2021), requires a predefined grouping into a discrete set of tasks. By contrast, our use of language to define tasks enables a more subtle and composable task space.

**Language-conditioned inverse RL**  Although a few prior works have used unrestricted natural language to define rewards for robotic manipulation tasks using either binary classification (Shao et al., 2021; Silva et al., 2021; Nair et al., 2022a) or contrastive vision-language alignment (Fan et al., 2022; Nair et al., 2022b; Ma et al., 2023a; Sontakke et al., 2023), these methods have not fully leveraged the temporal ordering of frames in their objective to encourage increasing scores over a successful episode. We instead propose to explicitly learn increasing rewards for partial trajectories making progress towards solving the task. Moreover, most prior methods use only a single image or a pair of images, whereas we consider the full episode to better represent partially observable tasks. Moreover, many prior works only considered data from the actor's own observation space (Silva et al., 2021; Fan et al., 2022; Nair et al., 2022a), whereas we propose to learn from cross-embodiment data allowing zero-shot transfer to a robot with different morphology, kinematics, and visual appearance.

Specifically, a few prior works (Fan et al., 2022; Baumli et al., 2023; Rocamonde et al., 2023) have explored CLIP-based models for learning vision-language reward functions. However, these works have been limited to image features of the current time step (Baumli et al., 2023; Rocamonde et al., 2023) or a snapshot of the most recent history (Fan et al., 2022). We instead propose to score behavior at the time series level by comparing the task description and a full video trajectory. Despite its name, RoboCLIP (Sontakke et al., 2023) (proposed to solve a similar problem to our work) is not based on the CLIP architecture or the pretrained representations unlike our method, but the S3D architecture (Xie et al., 2018), and mainly shares the InfoNCE loss with CLIP.

## 3 Video-Language Critic

We propose to learn language-conditioned robotic manipulation by first training an embodiment-agnostic reward function on video-caption pairs, and then using the learned reward model to guide the autonomous training of a robot-specific policy. To serve as a useful reward signal for downstream policy learning, the learned function should accurately represent the intended task, while providing enough signal to the agent to enable efficient learning (Ackley & Littman, 1992; Singh et al., 2009; 2010; Sorg, 2011). It should exhibit at least two key properties: *accuracy* and *temporal smoothness*. Making progress in the specified task should be rewarded with positive feedback with as little delay as possible, i.e., the function should smoothly increase over a successful execution. In fact, the problem of optimal reward shaping is equivalent to learning the value function for the optimal policy (Sorg, 2011), suggesting that an optimal densely shaped reward should monotonically increase over a successful demonstration (assuming the reward we ultimately wish to maximize corresponds to sparse goal reaching). Moreover, the end-of-episode scores for successful trajectories should exceed those of incomplete or failed executions: classification accuracy between successes and failures should be high. With these desiderata, we formulate Video-Language Critic, a language-conditioned reward model trained with cross-entropy and sequential ranking objectives to encourage progressively increasing scores over a successful video's duration.

**Contrastive video-language training**  Our approach is motivated by the success of contrastive image-language pretraining and the wide applicability of pretrained CLIP (Radford et al., 2021) encoders as foundation models. The problem of comparing observed behavior to a desired task description is analogous to the setting of CLIP; however, we extend the contrastive learning approach to scoring videos. Compared to a single image, using sequences of frames sampled across the full trajectory increases the generality of our reward function, and could allow it to handle non-Markovian (i.e., history-dependent) tasks. Such tasks might involve partial observability, repetitive or circular movements, or be described relative to an earlier state; even simple object displacement tasks may fall in this category.

**Reward model architecture**   We define video and text encoder networks similar to CLIP4Clip used for video retrieval (Luo et al., 2022), the task of finding videos within an existing dataset that most closely match a given textual query. The general architecture is shown in Fig. 1. First, each video frame is processed with an image encoder network while the video caption is processed with a text encoder, both initialized with CLIP in order to benefit from its large-scale vision-language pretraining. Luo et al. (2022) tested different aggregation strategies for reasoning over the resulting sequence of image features. In video retrieval, averaging image features over time was found to be sufficient, and no performance benefit could be obtained with an attention-based aggregation. While video retrieval shares similarities with our setting, task progress evaluation requires a much more nuanced understanding of temporal dynamics: for one, reversing the video should typically result in a very different reward value.

To support this, it is necessary for the temporal aggregation function to process the input frames as an ordered sequence, which in the case of Transformer aggregators is achieved using position embeddings. Furthermore, we find that embedding both modalities independently of each other and comparing with a cosine similarity, as done by CLIP as well as CLIP4Clip's *sequence Transformer* aggregation, causes the resulting video representation to lose too much of the information relevant to task completion. Thus, instead of using cosine similarity, **we train a single temporal aggregation Transformer to directly output a similarity score** based on the concatenated textual and image features. Our architecture is compared with the mean pooling and sequence Transformer variants in Figure 5. The text and the visual encoders' weights are fine-tuned and the aggregation function is trained from scratch for the video-text matching task. We refer to the full architecture, consisting of both the encoder networks and the aggregation Transformer, as a similarity function $S_\theta$, parameterized by $\theta$.

**Contrastive objective**   As training signal, we wish to leverage weak supervision from video-level captions without known spatial or temporal extent. We use a contrastive objective function to encourage each caption to better match its corresponding video than other videos, and vice versa. We therefore train the similarity prediction network $S_\theta$ with symmetric cross-entropy as done by Radford et al. (2021), i.e., with the mean of text-to-video and video-to-text cross-entropy terms, for video-caption pairs $(v^i, c^i), i = 1..N$:

$$\mathcal{L}_{xent} = (\mathcal{L}(v^{1:N}, c^{1:N}) + \mathcal{L}(c^{1:N}, v^{1:N}))/2, \tag{1}$$

with the cross-entropy loss from modality $x$ to modality $y$ defined as:

$$\mathcal{L}(x^{1:N}, y^{1:N}) = -\frac{1}{N} \sum_{i=1}^{N} \log \frac{\exp(S_\theta(x^i, y^i))}{\sum_{j=1}^{N} \exp(S_\theta(x^i, y^j))}. \tag{2}$$

**Sequential ranking objective**   Video inputs also contain implicit information about the relative *ranking* of states, which is not leveraged in prior reward learning approaches (Fan et al., 2022; Nair et al., 2022a;b; Karamcheti et al., 2023; Ma et al., 2023a; Sontakke et al., 2023).[2] We propose to learn from this temporal signal by extending the cross-entropy objective with a sequential ranking term. Each subsequent state in a successful trajectory should, in general, have higher value for completing the task than its predecessors, which the reward function should reflect. Our total loss then becomes:

$$\mathcal{L}_{VLC} = \mathcal{L}_{xent} + \frac{\alpha}{N} \sum_{i=1}^{N} \sum_{t=1}^{|v^i|-1} |S_\theta(v^i_{1:t}, c^i) - S_\theta(v^i_{1:t+1}, c^i)|_+ \tag{3}$$

where $\alpha$ is a hyperparameter balancing both objective terms and $|x|_+$ denotes $\max(x, 0)$.

In order to ensure the reward model learns to discriminate videos based on task completion rather than simply the presence of relevant objects in the scene, it may be beneficial to include failure examples featuring similar environments. Task failures are typically easier to generate than success examples and thus fairly inexpensive to collect. When available, we leave these videos uncaptioned and treat them only as additional negatives in contrastive learning, and do not include them in the ranking loss.

---

[2]While Ma et al. (2023a) use order information, they do not use relative ranking: consecutive frames are used bidirectionally as negatives in their contrastive objective, with their representations pushed apart from each other. This encourages the frames' representations to vary gradually over the video time span but does not leverage any prior on the later state having a higher value.

# 4    Experiments

We evaluate the accuracy and effectiveness of the learned video-language rewards on simulated robotic manipulation tasks from the Meta-World benchmark (Yu et al., 2019). We evaluate VLC's ability to inform successful policy training in three settings of increasing difficulty. First, we assess the ability of our model to jointly represent several robotic tasks with a single language-conditioned prediction network in Section 4.1. Second, we test our models' ability to generalize to unseen Meta-World tasks with the help of vision-language pretraining as well as extrapolation from training tasks in Section 4.2. In Section 4.3, we demonstrate our method's out-of-domain transfer ability: VLC is used to learn an embodiment-agnostic reward function for any language-conditioned manipulation task by observing a variety of robot actors from Open X-Embodiment (Open X-Embodiment Collaboration et al., 2023), a large dataset collected from a variety of real-world robots in different environments. We further report comparisons to prior work, both quantitatively and qualitatively, in Section 4.4, and demonstrate VLC's effectiveness in planning with a known dynamical model in Section 4.5. Finally, in Appendix F, we investigate the model's robustness to visual variations (brightness shift and image noise).

**Training details**   We perform hyperparameter selection, ablation studies, and finalize all training details on data from VLMbench (Zheng et al., 2022), a second robotic benchmark. This avoids Meta-World specific tuning and allows for a fairer comparison with prior work. Details on this dataset and our model ablations are included in Appendix A. We observe the same hyperparameters perform well both on VLMbench videos and in interactive RL policy training on Meta-World, which highlights the method's applicability across domains. Further VLC training details are reported in Appendix C. To train control policies, we use Soft Actor-Critic (SAC) (Haarnoja et al., 2018), an off-policy RL algorithm. As the reward function, we use either a sparse task completion reward, equal to 1 if the task has been solved and 0 otherwise, or a weighted sum of the learned VLC reward and the sparse reward. As the similarity $S_\theta(v_{1:t}, c)$ predicted by VLC can take arbitrary values, we apply a few normalization operations to aid in the stability of RL optimization (van Hasselt et al., 2016) (details in Appendix C.2).

**Environments**   We evaluate VLC in RL policy training on robotic manipulation tasks from the Meta-World benchmark. As the focus of this work is the problem of learning a foundation reward model, we keep to a standard single-task policy training setup and condition on full state information as defined by Meta-World (see Appendix D for more details on the environment definitions). Compared to using image-only observations, this reduces training time and computational cost while allowing us to demonstrate the effectiveness of our reward function. We consider the subset of tasks that can be reliably solved using the dense Meta-World rewards, which are manually specified for each environment based on the full state of the environment. Specifically, we include tasks that can be solved with $\geq 98\%$ success within a maximum training length of 800,000 steps. We further split these in half into 12 easy (learned in $<$240,000 steps) and 13 hard tasks (240k – 800k steps).

## 4.1    Multi-task reward function

To validate VLC's effectiveness as a multi-task reward function, we first train our model on video data from all 50 tasks. We collect 40 video demonstrations per task for a total video dataset of 2000 successful executions. We further collect 1600 failure examples by replacing the demonstrator's actions with random actions with probability 0.7, and refer to this joint dataset as MW50 (short for Meta-World). We do not make any modifications to the data generating process to explicitly encourage exploration, as we want to validate our method in the context of existing offline data, which typically does not cover the full state space. A key challenge VLC needs to overcome is to sufficiently generalize from the successes and failures present in the data to evaluate out-of-distribution trajectories, as the RL policy may act very differently from the demonstration data.

The policy training results are shown in Table 1, with learning curves for the hard task set in Fig. 2. To summarize learning speed with a single number, we report success rate of the policy evaluated at the training length at which the manually specified Meta-World reward solves the task to $\geq 98\%$ success. VLC trained on MW50 enables improved sample efficiency relative to the sparse reward only, which demonstrates that VLC

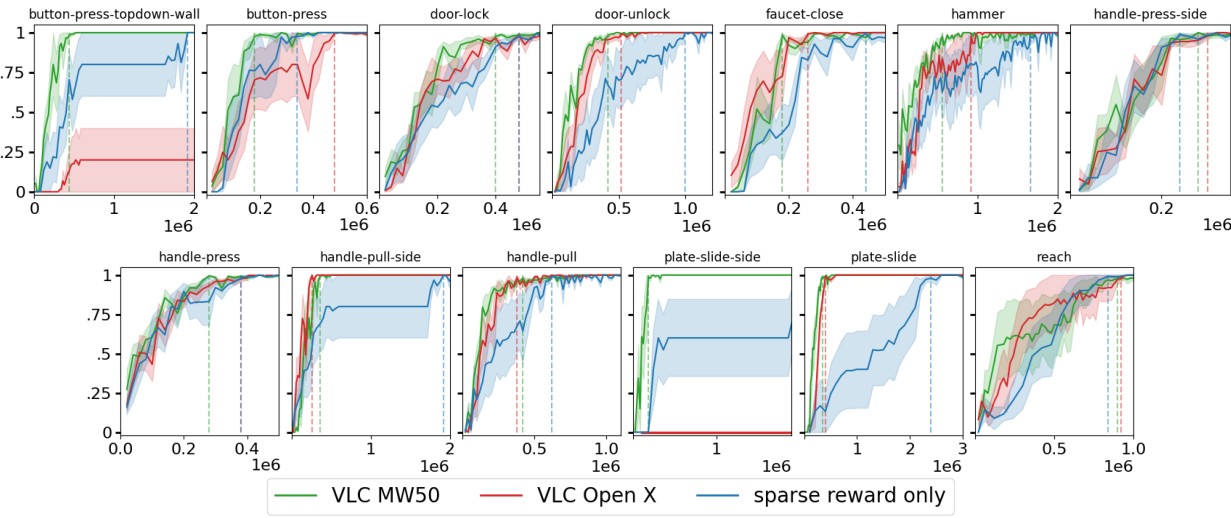

Figure 2: **Success rate of RL policies on hard Meta-World tasks**, over the number of environment steps (mean of 5 random seeds and standard error). Dashed lines denote convergence ($\geq 0.98$).

Table 1: Success rates (%) of sparse reward only training vs. VLC trained on either MW50 or Open X, evaluated at the training length at which the Meta-World hand-designed reward solves the task.

|  | sparse only | VLC MW50 | VLC Open X |
|---|---|---|---|
| easy (12) | $53 \pm 10$ | $\mathbf{65} \pm 9$ | $\mathbf{56} \pm 11$ |
| hard (13) | $62 \pm 8$ | $\mathbf{91} \pm 4$ | $\mathbf{73} \pm 9$ |
| mean | $58 \pm 6$ | $\mathbf{78} \pm 5$ | $\mathbf{65} \pm 7$ |

can sufficiently generalize to trajectories not seen in demonstration data, and can effectively represent task progress for multiple tasks at once. However, a few tasks, such as Handle Press, are learned in so few trials even with sparse reward alone that there is little room for improvement in reward design, and learning is instead bottlenecked by the policy training's sample efficiency. This is why the biggest gains are obtained for the harder tasks.

## 4.2 Task generalization to unseen environments

Next, we evaluate VLC's ability to generalize to entirely unseen tasks using language conditioning. For this purpose, we split Meta-World into 40 training and 10 test tasks (every 5th task alphabetically). This leaves roughly 1600 successful and 1300 unsuccessful videos as training data – we refer to this subset as MW40. Of the test tasks, 2 are in the easy set, 4 in hard and 4 are unsolved even with the curated single-task Meta-World rewards, and hence not our primary evaluation target.

Success rates of RL policies on these 6 held-out tasks are shown in Table 2 (calculated similarly to Table 1), with learning curves in Fig. 3. In most tasks, the addition of VLC rewards improves sample efficiency of RL training despite the tasks having never been seen in reward function training, with average success rates 31 percentage points over the sparse baseline and an average sample efficiency improvement of over 3x.

In addition to task generalization, we further evaluate VLC in out-of-distribution visual conditions in Appendix F.

## 4.3 Embodiment generalization to unseen domains

The advantage of our method, and pretraining a reward function in general, is that no data collection on the target robot and in the target environment is required. To demonstrate this, we train VLC on

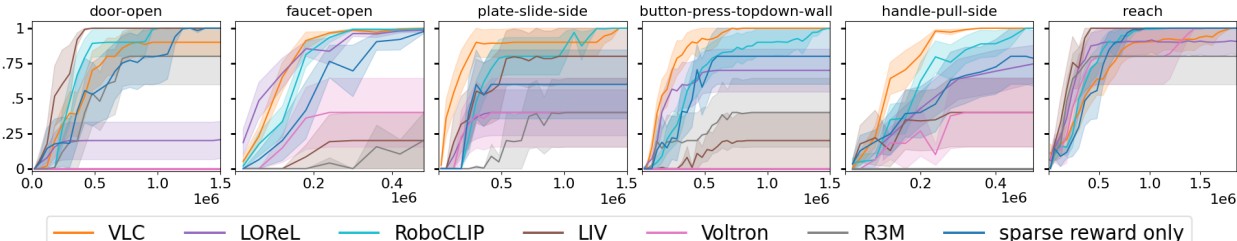

Figure 3: **Success rate of RL policies on held-out Meta-World tasks**, over the number of environment steps. VLC achieves improved sample efficiency compared to any prior approach in 3/6 tasks.

Table 2: Success rates (%) of VLC and prior work trained on MW40, evaluated on 6 unseen tasks (2 easy, 4 hard) at the training length at which the Meta-World hand-designed reward solves the task.

|          | LOReL      | RoboCLIP   | LIV        | Voltron    | R3M        | VLC (ours)     | sparse only |
|----------|------------|------------|------------|------------|------------|----------------|-------------|
| easy (2) | 56 ± 37    | 46 ± 45    | 30 ± 18    | 18 ± 18    | 2 ± 2      | **57** ± 37    | 39 ± 25     |
| hard (4) | 59 ± 8     | 51 ± 14    | 44 ± 19    | 35 ± 14    | 24 ± 18    | **80** ± 10    | 42 ± 8      |
| mean     | 58 ± 11    | 49 ± 15    | 39 ± 13    | 29 ± 11    | 16 ± 12    | **72** ± 13    | 41 ± 8      |

cross-embodiment data from Open X-Embodiment (Open X-Embodiment Collaboration et al., 2023). We use the language-annotated subset, with a total of 698,000 episodes of diverse tasks filmed in various real-world robotic labs. Although some of this data does feature the Sawyer robot used in Meta-World simulations, this is only a marginal subset of 0.33% of the language-annotated videos. Moreover, the domain gap remains significant due to real-world variations in objects, backgrounds, lighting conditions, task instances and instruction formats, as well as the embodiment gap between the simulated and the real robots.

We successfully train policies (see Table 1 and Fig. 2) using Open X trained models despite a significant domain gap, highlighting the generalizability of large-scale vision-language training. We obtain an average 2x sample efficiency gain over the sparse reward in tasks solved by both rewards (sample efficiency is ill-defined if either does not solve the task), with particularly large improvements in Handle Pull Side (7x), Reach Wall (7x) and Slide Plate (6x), and an average 7 percentage point success rate increase across all 25 tasks despite misrepresenting a few tasks. Note that unlike RT-X (Open X-Embodiment Collaboration et al., 2023), our method does not use action labels, and remains equally applicable on observation-only data.

## 4.4 Comparison to prior work

To validate VLC's benefits, we compare its performance to prior language-conditioned reward models LOReL (Nair et al., 2022a), RoboCLIP (Sontakke et al., 2023), LIV (Ma et al., 2023a), Voltron (Karamcheti et al., 2023) and R3M (Nair et al., 2022b), each fine-tuned on MW40. A breakdown of the key differences between VLC and these baseline methods is shown in Table 6 in Appendix E; in summary: VLC is the only method to use a sequence ranking objective or a temporal aggregation Transformer, and one of only two to use history conditioning. For the LOReL baseline, we used the proposed binary classification objective, reversed negatives and 2-frame conditioning (first and last), while keeping the architecture and CLIP pretraining identical to our method. This is to ensure LOReL's smaller and older original architecture and lack of visual representation pretraining did not account for any difference in performance. The original implementations were used for all other methods. Further training and implementation details for the baselines are also included in the Appendix E.

We find VLC's combination of cross-entropy and the sequential ranking objective, temporal Transformer architecture as well as full video conditioning to produce more informative reward predictions than existing methods, as shown by faster policy training on average in Table 2 and Fig. 3. For increased statistical significance, we use 5 additional random seeds (10 total) for VLC and the two strongest baselines: RoboCLIP and LOReL. Moreover, on qualitative inspection of the shape of the predicted rewards (Appendix G), we find

VLC's outputs to better distinguish successes from failures compared to either RoboCLIP or LOReL. Thanks to its conditioning on more frames of execution history and the sequential loss term, VLC also produces rewards that more smoothly increase over successful episodes than either prior method.

### 4.5 Model-based evaluation

Table 3: **Model-based experiment:** Accuracy (%) of VLC-MW40 in identifying a successful execution out of 6 sampled trajectories, averaged over 50 random scene initializations for each held-out task. Two action primitives are required to solve Assembly and Pick Out of Hole: first grasping the required object, then moving it as specified, such as lifting it out of the hole.

| | Plate Slide Side | Stick Push | Handle Pull Side | Faucet Open | Door Open | Coffee Push | Button Topdown Wall | Reach | Pick Out of Hole (Grasp) | Pick Out of Hole (Goal) | Assembly (Grasp) | Assembly (Goal) | Mean |
|---|---|---|---|---|---|---|---|---|---|---|---|---|---|
| VLC | 100 | 20 | 96 | 100 | 40 | 94 | 54 | 82 | 100 | 100 | 97 | 100 | **80** |
| RoboCLIP | 88 | 60 | 58 | 4 | 86 | 32 | 4 | 14 | 26 | 82 | 44 | 50 | 46 |
| LOReL | 26 | 14 | 44 | 14 | 4 | 22 | 22 | 28 | 72 | 54 | 100 | 30 | 36 |

As a pretrained reward function, VLC can also inform model-based planning. We demonstrate this in a proof-of-concept experiment, where we do not learn the model but instead assume access to a known transition model as well as action primitives. The action primitives include grasping and reaching, parameterized by target positions (such as the locations of objects detected in the scene), and are defined using segments from the expert policies available in Meta-World. We evaluate VLC's ability to identify the action primitive with the correct execution for a held-out task by assigning it a higher score than for incorrect executions. In each task, we compare one successful trajectory with 5 unsuccessful ones with randomly sampled target positions.

We find VLC to generalize well to these tasks zero-shot, with 80% mean accuracy on held-out tasks, outperforming RoboCLIP (46%) and LOReL (36%). A breakdown per task is shown in Table 3.

## 5  Limitations and future work

**Suboptimal demonstrations**   Our sequence ranking objective most benefits from optimal or high-quality trajectories. However, this is also true for imitation learning approaches, and learning a reward function (unlike imitation learning) still allows the policy to outperform the existing trajectories by executing the task faster or by finding shortcuts: the highest rewards are still assigned to the final states, so getting to one faster is encouraged. Moreover, our objective does not impose consistent increase, only non-decrease of rewards. Therefore, a suboptimal section of a demonstration can be assigned a constant reward, representing non-progress in the task, and the objective only assumes that the suboptimal parts do not make negative progress in the task, which is a much less restrictive assumption.

**Choice of evaluation benchmarks**   In our formulation, we focused on tasks whose rewards can be simplified to goal reaching, but other rewards may be relevant (such as penalizing large actions). We further assumed that a sparse task completion reward can be provided, which may not always be the case.

In this work, we focused on simulated robotic manipulation tasks, and evaluation on a real robot is left for further work. Moreover, extending VLC to learn from videos of humans performing manipulation tasks, such as from the Something-Something dataset (Goyal et al., 2017), is a promising avenue for pushing the generalization capabilities of pretrained reward functions in the future.

# 6 Conclusion

We proposed Video-Language Critic (VLC), a method for training a foundation reward model for vision-language manipulation. In particular, we train our critic using contrastive video-language alignment and a ranking loss encouraging monotonic increases for successful trajectories. Our model predicts a language-conditioned state-value function conditioned on only a history of image observations, and can therefore readily be scaled to leverage external observation data from other actors. VLC can be used for various downstream tasks, such as model-free and model-based reinforcement learning. Further, we experimentally validated its usefulness as a reward function in out-of-distribution tasks (unseen, held-out environments in the same domain) and out-of-domain tasks (an unseen environment and embodiment in a different domain). Our experiments on Meta-World demonstrated improved results compared to 5 prior methods and a sparse reward baseline, with success rate increases of 14 and 31 percentage units, respectively, and sample efficiency gains of 30% and 300%, respectively. Unlike methods based on predicting expert actions in a given embodiment, VLC remains applicable in the absence of action labels. Therefore, it can be extended to equally learn from videos of humans in future work.

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

## Supplementary Material

## A    Model ablations on VLMbench

### A.1    VLMbench dataset

We use the VLMbench manipulation task suite to develop and validate our method without any Meta-World specific tuning. For this purpose, we collect 2700 video demonstrations and 1600 failure cases from variations of the picking task – covering different object shapes, sizes, colors and relative positions, as well as distractor objects. The natural language instructions match this diversity in task variants, such as *Grasp the cylinder* or *Grasp the cyan object*, and require distinguishing relevant objects from distractors with either absolute (color, shape) or relative (size, position) properties, such as the the *larger* or the *front* object. For more details on the benchmark, see Zheng et al. (2022).

We use these VLMbench videos to validate VLC design decisions, but defining a single informative metric on the dataset of video-caption pairs $(v^i, c^i)$, $i = 1, ..., N$, is difficult. Test loss, video retrieval metrics such as mean recall, or classification metrics such as area under the ROC curve do not correspond well to the models' ability to model task progress. The main difficulty is that part of the caption-to-video matching task can be solved by simply connecting objects referred to in the caption to objects present in the scene, without considering temporal information or actual task success.

To support informative evaluation of our models, we therefore further define a set of 19 test episodes: in each test case, the same initialization of the scene is used to generate alternative trajectories that grasp at different objects in the scene, only one of which solves the correct task. The accuracy over this set of videos is our main model selection metric of interest, i.e., in how many out of 19 instances does the model assign a higher score to the successful video than any incorrect video from the same initialization. Out of evaluation metrics available at training time, we find video-to-text cross-entropy to correlate the most with this test-time accuracy, and so use this metric on a set of validation trajectories to choose model checkpoints.

### A.2    Ablation results

We compare two temporal aggregation methods as proposed by Luo et al. (2022): the sequence transformer and the tight-type transformer. The sequence transformer aggregates the sequence of image features $[\text{ViT}(v_1), \text{ViT}(v_2), ..., \text{ViT}(v_T)]$ into a single embedding vector $S_\theta(v_{1:T})$, which it then compares to the caption embedding $\text{TextEnc}(c)$ with cosine similarity. The tight-type transformer, on the other hand, includes the caption embedding as an additional input to the temporal aggregator $S_\theta(v_{1:T}, c)$, as shown in Fig. 1.

We report the results of our ablation study in Table 4. In addition to the choice of architecture, we observe performance gains from adding image augmentations from the Albumentations library (Buslaev et al., 2020), by sampling frames randomly from uniform intervals instead of deterministic uniform sampling, the addition of the sequence ranking term, as well as considering failure examples only as negatives in the contrastive objective, and report results using these settings in Section 4.

## B    Model ablations on Meta-World

In order to isolate the contributions of the VLC architecture and of the sequence ranking objective to the observed performance improvements on Meta-World, we run an additional ablation using only the contrastive component of the loss (by setting $\alpha = 0$). Moreover, as a second ablation variant, we use our architecture together with the LIV objective.

The resulting training curves are shown in Figure 4, with the success rates included in Table 5. All policy training runs were repeated with 10 random seeds. The full VLC objective outperforms both ablations on average across the 6 Meta-World held-out tasks – by 5 and by 12 percentage points, respectively. Although the no-sequence-ranking baseline obtained the highest success rate on the easy tasks, as there are only two tasks in this set, results on the intermediate set and the average across all tasks are more meaningful. The gains over the no-sequence-ranking ablation are small but consistent with the gains observed on VLMbench

Table 4: Accuracy on VLMbench test episodes for various model ablations. $\alpha$ is the weight of the sequence ranking loss term.

| Architecture | $\alpha$ | Data augmentations | Failures | Accuracy (%) |
|---|---|---|---|---|
| Sequence transf. | 0 | - | as negatives only | $51.6 \pm 3.1$ |
| " | " | image | " | $62.1 \pm 4.2$ |
| " | " | frame sampling | " | $44.2 \pm 3.6$ |
| " | " | image & frame sampling | " | $60.0 \pm 4.3$ |
| " | " | image & frame sampling | yes | $72.6 \pm 7.1$ |
| Tight type | " | - | as negatives only | $52.6 \pm 4.7$ |
| " | " | image | " | $72.6 \pm 3.1$ |
| " | " | frame sampling | " | $55.8 \pm 8.4$ |
| " | " | image & frame sampling | " | $80.0 \pm 4.5$ |
| " | " | image & frame sampling | yes | $76.8 \pm 4.6$ |
| " | 3.3 | image & frame sampling | as negatives only | $82.1 \pm 4.3$ |
| " | 10 | " | " | $83.2 \pm 2.0$ |
| " | 33 | " | " | $\mathbf{88.4} \pm 5.4$ |
| " | 100 | " | " | $85.3 \pm 3.1$ |

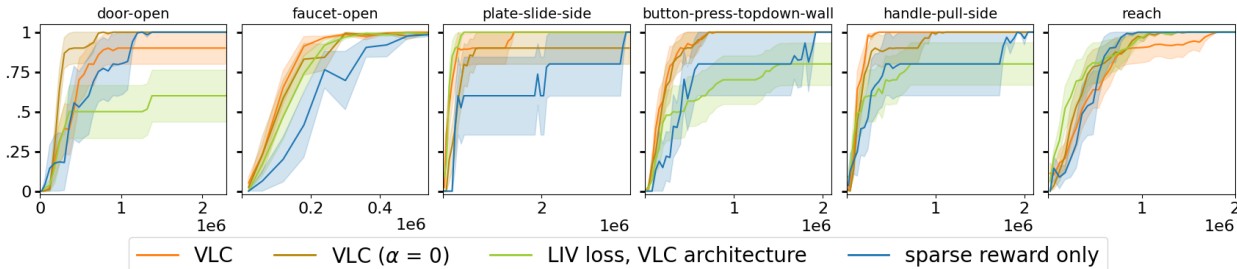

Figure 4: Ablations on the choice of objective on held-out Meta-World tasks, compared to the full VLC objective. $\alpha = 0$ refers to only using the contrastive objective, while the LIV loss is a combination of the contrastive objective and the VIP (value-implicit pretraining; (Ma et al., 2023b)) loss applied on the image representations. Both ablations use the VLC architecture conditioned on 12 frames.

Table 5: Success rates (%) of VLC and ablations trained on MW40, evaluated on 6 unseen tasks (2 easy, 4 hard) at the training length at which the Meta-World hand-designed reward solves the task.

| | VLC | VLC ($\alpha = 0$) | LIV loss, VLC archit. | sparse only |
|---|---|---|---|---|
| easy (2) | $57 \pm 37$ | $\mathbf{70} \pm 23$ | $46 \pm 35$ | $39 \pm 25$ |
| hard (4) | $\mathbf{80} \pm 10$ | $66 \pm 9$ | $67 \pm 9$ | $42 \pm 8$ |
| mean | $\mathbf{72} \pm 13$ | $67 \pm 8$ | $60 \pm 12$ | $41 \pm 8$ |

data, our second evaluation benchmark (see Appendix A). Furthermore, it is possible that tuning the $\alpha$ parameter on Meta-World would improve performance, but we did not explore this and instead tuned all hyperparameters on VLMbench only.

Note that applying the LIV loss to our architecture requires the time-contrastive loss on images to be applied after the visual encoder but before the temporal aggregator, whereas the vision-language contrastive loss is applied to the final output.

## C   Training details

### C.1   Reward training

We subsample the videos to 12 time steps. Capping the maximum video length is a practical choice both in terms of learning ability and computational cost. We keep the default value of 12 frames in CLIP4Clip, though we change these to be linearly sampled from across the entire video. Informed by the findings of our ablation studies in Section A, at training time, we additionally apply image augmentations and randomize frame sampling. We set $\alpha$, the ranking loss weight, to 33 based on accuracy on VLMbench test episodes.

**Computational cost**   Reward training on Meta-World videos took 2 hours for MW50 on a single NVIDIA A100 GPU, and 1 hour for MW40 on a GeForce RTX 3090 GPU. Training on the significantly larger Open X-Embodiment dataset took 256 hours (nearly 11 days) on a single A100. However, we believe this length could be greatly reduced in future work by improving data loading throughput and running on multiple GPUs.

### C.2   Policy training

For RL training experiments, we adapt the SAC implementation of CleanRL (Huang et al., 2022). Policy evaluation is done every 20,000 timesteps for 50 episodes. Both the actor and critic networks contain three hidden layers of size 400, and optimization is done using Adam (Kingma & Ba, 2017). Other algorithm hyperparameters were kept at the implementation's default values.

**Reward normalization**   As reward model predictions for the starting state can vary across initial states, even for the same task, we shift the rewards of the episode such that $r_1 = 0$, as the subsequent behavior should be scored relative to this state. In addition to this, we apply the standard normalization logic from Gymnasium (Towers et al., 2023), which scales reward values such that their exponential moving average has fixed variance $(1 - \gamma)^2$.

**Reward component weighting**   We experimentally set the relative weights of the VLC and sparse reward components to 1 and 50, respectively, the motivation being that the sparse reward, once obtained, should be able to override the dense intermediate reward predictions. We chose these values after testing three other settings: (0.01, 10), (0.1, 10), and (0.1, 20), which also performed quite well. For sparse reward only experiments, we did not find significant differences in the scale of the reward, but for some tasks using a weight of 50 seemed to perform better than 1 or 10, so we report results using this value for consistency with the VLC experiments.

**Computational cost**   Training length and hence computational cost varies considerably across tasks. We terminate training after convergence to $\geq 98\%$ success (averaged over the 10 most recent evaluations), or after a maximal training length set per task. The resulting average training length across all 25 tasks was 600k environment steps for VLC MW50 and 750k for the Open X trained model. The corresponding GPU hours vary slightly based on exact architecture used, but we obtained approximately 750k steps in 24 hours on a single NVIDIA V100 or P100 GPU. Our total computational budget was therefore 19.2 GPU hours $\times$ 5 random seeds $\times$ 25 tasks = 2,400 GPU hours for the RL training experiments with VLC MW50 and 24 $\times$ 5 $\times$ 25 = 3,000 hours with VLC Open X. For the MW40 task generalization experiments, the average training length was again 600k for VLC, but 10 random seeds were used, so the corresponding cost for training on 6 tasks was approximately 1150 GPU hours for our method, and slightly more for baselines that took longer to train. Running inference on the VLC architecture to predict the reward for one time step (with batch size 1) takes up an additional 800 MB of GPU RAM, and 11 ms on a H100 GPU or 29 ms on a V100.

## D   Meta-World environment definitions

The Meta-World environments are a set of continuous control robotic manipulation tasks. The state space of each environment is $\mathbb{R}^{39}$, containing the positions and orientations of relevant objects as well as the xyz-

position of the robot's end-effector. An action $a$ in $[-1, 1]^4$ consists of the desired end-effector xyz-translation, and one dimension for controlling the openness of the gripper. We refer to Yu et al. (2019) for more details on the state information.

As task captions, we use the first sentences of the environment descriptions included in Table 2 in Appendix A - Task Descriptions in Yu et al. (2019). These are fairly succinct and intuitive (examples: *Rotate the faucet counter-clockwise*, *Press a button from the top*, *Pull a handle up*) and we did not need to modify them further.

## E   Baseline implementations

For the LOReL baseline experiment, we used the proposed binary classification objective, reversed negatives and 2-frame conditioning (first and last), while keeping all of our other training details and data augmentations identical to our method. This is to ensure LOReL's smaller and older original architecture as well as lack of visual representation pretraining did not account for any difference in performance. The original implementations were used for all other methods, and pretrained checkpoints (e.g., HowTo100M (Miech et al., 2019) pretraining for RoboCLIP) were reused when applicable. Pretraining datasets, architectures, training objectives and history lengths for each model are listed in Table 6. The difference between temporal aggregation and cosine (or Euclidean) based architectures is illustrated in Figure 5.

Although not considered as a source of data in the original works, we also use our failure videos as additional negatives in the contrastive objective of LIV, RoboCLIP and R3M. Model selection was performed based on validation loss on held-out trajectories. Otherwise, the training procedure and hyperparameter settings were kept unchanged from the original works. In policy training, we apply identical reward normalization (offset and scale, as described in Section 4) for all methods, and reuse the multiplier 50 for the sparse reward.

## F   Robustness to visual variations

To evaluate VLC's generalization to visual observation shift and noise, we run additional experiments where the brightness of the scene is reduced, and uniformly sampled pixel noise is added per image. Example images of this perturbation compared to the original appearance are included in Fig. 9. The success rate curves are included in Fig. 7. 2/6 tasks' performance is somewhat affected by this noise, but 4/6 are unaffected.

Table 6: **Key differences to prior work**. Previous methods use various objectives, but VLC is the first to include a sequence ranking loss, encouraging smooth, dense rewards. Moreover, our model's conditioning on 12 frames instead of just 1 or 2 like each prior work apart from RoboCLIP allows history-aware progress evaluation, which is important for a variety of tasks (such as distinguishing *Turn faucet clockwise* and *Turn faucet counter-clockwise*). Finally, VLC defines rewards based on the output of the temporal Transformer, which is more expressive than compressing the text and video separately into an embedding space and comparing them with either cosine or Euclidean distance, as done by all prior works other than LOReL. As LOReL did not originally use any visual pretraining and used a small architecture, for a fair comparison, we compared to a reimplementation of its training objective while reusing CLIP pretraining and our architecture.

| | Objective | Num. images | Architecture | Reward defined by | Pretraining data |
|---|---|---|---|---|---|
| **VLC (ours)** | CLIP + **sequence ranking** | **12** | CLIP + temporal Transformer | **model output (temporal Transformer)** | CLIP |
| **LOReL** (original work) | binary classification | 2 | 12-layer CNN + small classifier & DistilBERT (Sanh, 2019) | **model output** | language only |
| **LOReL** (our reimplementation) | binary classification | 2 | CLIP + temporal Transformer | **model output (temporal Transformer)** | CLIP |
| **Robo-CLIP** | CLIP | **32** | S3D (Xie et al., 2018) | cosine | Howto100M (Miech et al., 2019) |
| **LIV** | CLIP + VIP (variant of time-contrastive; Ma et al. (2023b)) | 1 | CLIP (ResNet-based) | cosine | CLIP + EPIC-KITCHENS (Damen et al., 2018) |
| **Voltron** | visual reconstruction + language generation | 2 | Transformer encoder-decoder & DistilBERT | Euclidean | Something-Something v2 (Goyal et al., 2017) |
| **R3M** | CLIP + time-contrastive (Sermanet et al., 2018) + L1 sparsity | 1 | ResNet (He et al., 2016) & DistilBERT | Euclidean | Ego4D (Grauman et al., 2022) |

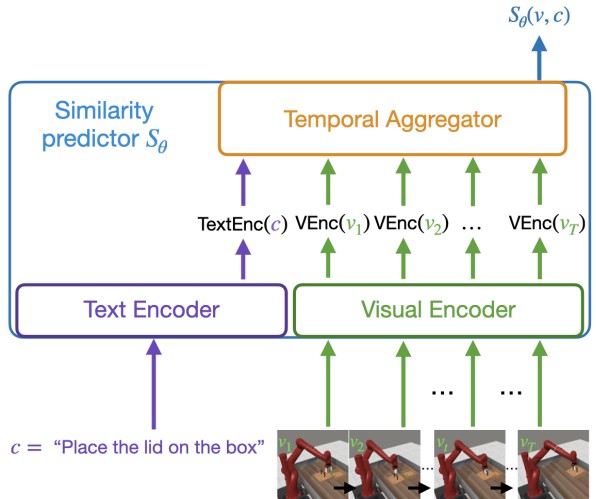

(a) VLC architecture: temporal aggregation is performed based on both the caption embedding and the sequence of image embeddings.

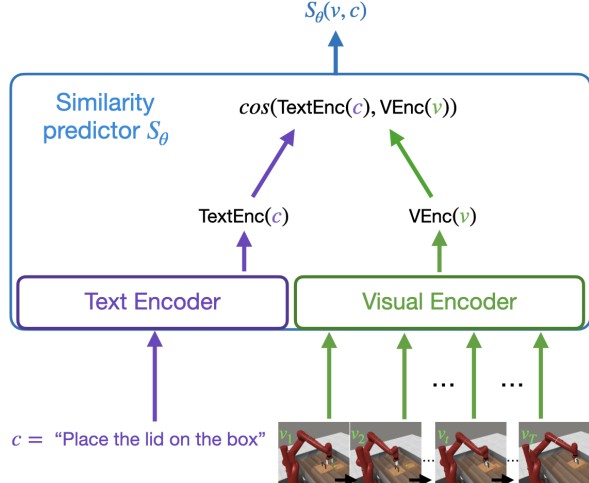

(b) Distance metric based aggregation: all temporal information is compressed into a fixed-size embedding before being compared with the caption embedding. In addition to the *cosine* and *Euclidean* based methods listed in Table 6, the *sequence Transformer* of Luo et al. (2022) also falls in this category.

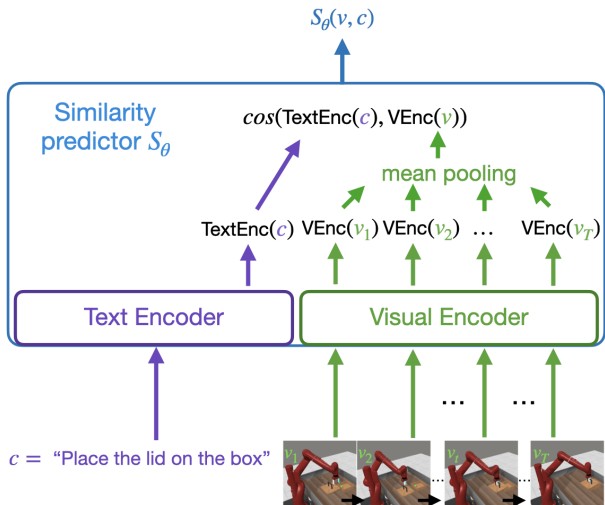

(c) Architecture with mean pooling temporal aggregation. Proposed for video retrieval in Luo et al. (2022), but not well suited to reward modeling due to order invariance.

Figure 5: Schematics of the differences between our temporal aggregation architecture (a), architectures that apply a predefined distance metric on top of separate embeddings of the text and the video (b; illustrated with cosine distance, but Euclidean distance has also been used), and mean pooling based architectures (c). Regardless of the choice of text and visual encoders, the main bottleneck in expressiveness in the architectures depicted in b), commonly used in vision-language reward models, comes from compressing the video into a single visual encoding with fixed dimensionality before conditioning on the task caption against which the video should be evaluated, whereas in our architecture, frame-wise information is preserved until the final aggregation step, which is also conditioned on the task caption.

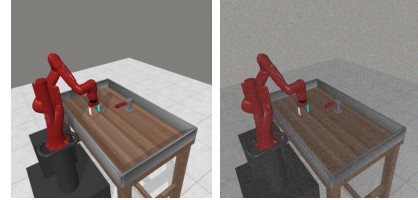

Figure 6: **Left**: Example observation from the Open Faucet task with the original brightness (0.3) and without image noise. **Right**: Example observation from the same scene with shifted brightness (0.01) and pixel-wise noise sampled uniformly from [0, 0.2].

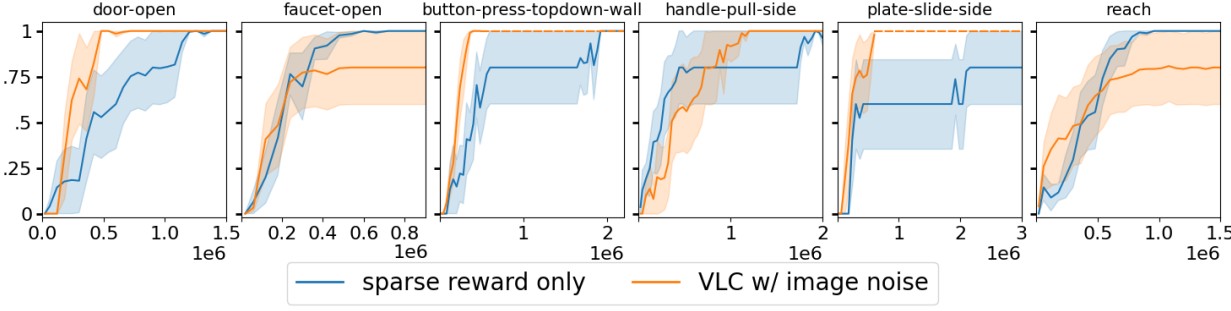

Figure 7: Success rates in VLC training under brightness shift and image noise (as shown in Fig. 9).

## G Qualitative evaluation

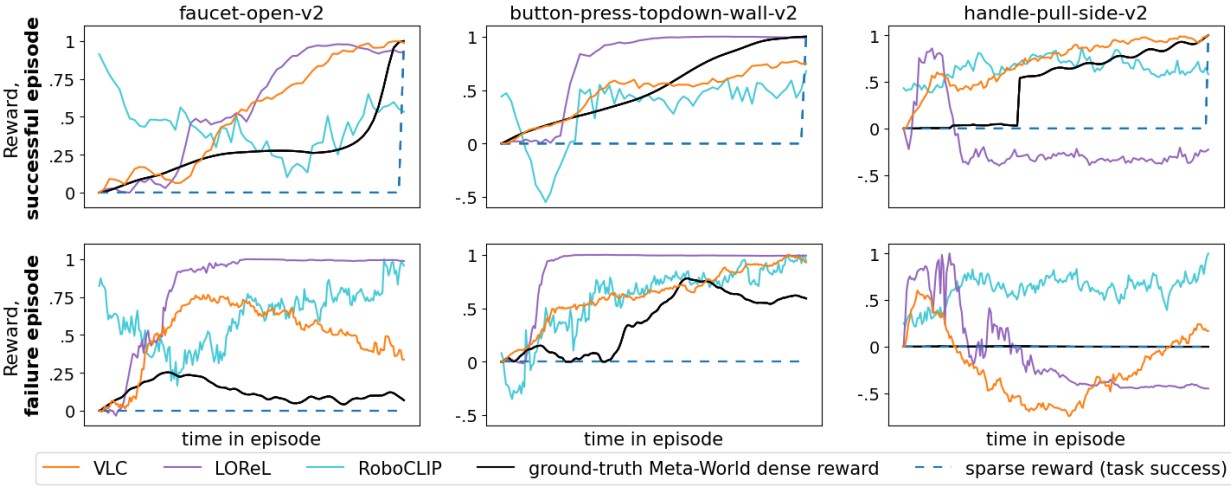

Figure 8: **Qualitative evaluation** of rewards predicted over time steps of an example successful (above) and unsuccessful (below) test episode. Predictions are offset so the episode start has reward 0, as done in policy training, except for RoboCLIP, which only assigns a reward for the final time step in RL training, as in the original work (the full curve is shown here for visualization only). In this figure, rewards are further normalized per task and method so that the success and failure episode rewards are comparable: e.g., in Faucet Open, VLC assigns at most 75% of its reward prediction for the end of the success episode to any step of the failure episode; this is similar in spirit to the scale normalization used in policy training (which instead normalizes running statistics). A good reward model gives higher rewards in the top row than the bottom row. Good correlation with the Meta-World reward also implies an understanding of the task, but is not strictly required for successful RL training.

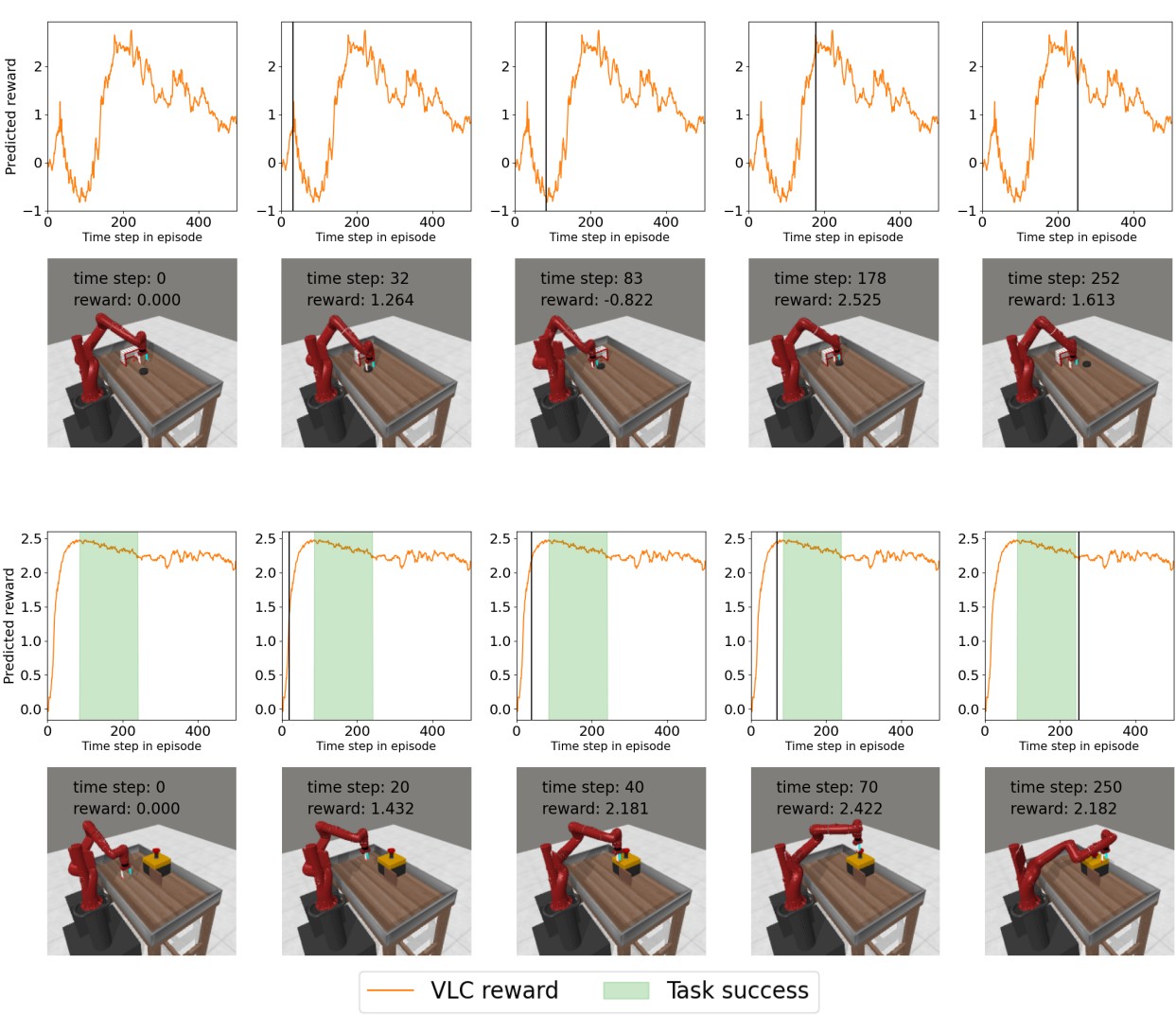

Figure 9: **VLC MW40 reward predictions** overlaid with example episodes from the Plate Slide Side (above; caption: *Slide a plate into a cabinet sideways*) and Button Press Top-down Wall tasks (below; caption: *Bypass a wall and press a button from the top*). The corresponding videos are included on the project website: https://sites.google.com/view/video-language-critic.

