# OpenReview forum: "Video-Language Critic: Transferable Reward Functions for Language-Conditioned Robotics"
_TMLR — Accepted by TMLR_

### Review · Reviewer_GSa5 · 2024-11-25

**Summary Of Contributions:**

The paper questions whether pre-training a reward model based on large datasets of video observations and language descriptions can improve performance and sample efficiency in environments where the original reward signal is often sparse. The alternative reward model is trained with contrastive learning over videos-captions pairs, and tested on the MetaWorld benchmark, demonstrating significant sample-efficiency gains and generalization capabilities.

**Audience:**

Yes

**Claims And Evidence:**

Yes

**Requested Changes:**

* Question: does the model require the full video trajectory from the beginning, or can it work using only a recent part of the trajectory, e.g., the last 5 frames?
* Question: To my understanding, the pre-trained reward model on MW50 is trained on successful trajectories, and then this model is used to provide reward for the same tasks you are training SAC on? So, in order to train on some task, you need to provide the reward model with a solution? Please clarify what is the point of that (using a reward model trained on solutions from the same tasks you are trying to solve). I would find the “VLC Open-X” more representative for real-world applications in this case.
* Question: Instead of training the reward on robotic data, do the authors think it is possible to train it on video data of humans performing tasks (e.g., Meta’s Ego4D dataset) and use it to guide robotic learning with the proposed approach? I believe this could be an interesting future direction and would like the authors’ thoughts on this. (non-critical)
* Stability: can the authors share details about the stability of the pre-training, as balancing between the cross-entropy and the ranking loss might be brittle (due to different scales of the losses or collapses of the ranking loss)? How consistent are the results across training runs (i.e., two runs with the same hyper-parameters result in the same performance?)
* Open -source code: do the authors plan to provide an open-source implementation and pre-trained models?
* Limitations and future work: please add a discussion of limitations and optionally some future work directions that arise from this work (I suggested one above).
* See “Weaknesses” and “Minor”.

**Strengths And Weaknesses:**

**Strengths**:
* Interesting problem and clear motivation.
* The proposed approach is novel for considering video and trajectory ranking as a reward model.
* Improved sample efficiency compared to (sparse) rewards from the environment.
* Generalization capabilities to unseen tasks.
* Detailed appendix.

**Weaknesses**:
* Architecture differences: in page 5, first paragraph, the authors try to convey in words the difference in their aggregator compared to previous works. I found it hard to understand the differences. A more detailed description, or even an illustration, would help to distinguish this work from previous work (can be in the appendix), in addition to Table 5, which is great.
* Limited evaluation environments: the method is only evaluated on the simulated MetaWorld.
* Language descriptions: to my understanding, manual crafting of prompts for each new task and environment is required. Please correct me if I’m wrong. If I’m correct, a discussion of how these language descriptions are created is missing (or it is somewhere in the paper and I missed it).
* While the sample efficiency is improved, training the downstream policy requires forwarding a trajectory through the large model at each step, which is more expensive than just using the reward from the environment. What is the cost of that (time and memory)? A discussion of that would be beneficial, especially if it is a limitation of the model.
* Representative rollouts: this work is strongly reliant on videos. I would have appreciated more examples of captions and observations (appendix) and representative successful/failure rollouts (supplementary material or webpage). I think this can emphasize the paper’s contribution.

**Minor**:
* Figure 2: The first graph’s title (“button-press-topdown-wall”) has a small alignment problem where the title overlaps with the next graph. Also occurs in Figure 3, but less severe.
* Related work, first paragraph: “without ranking existing trajectories or attempting to outperform prior data” - at this point of reading the paper, it is unclear what is “trajectory ranking” (ranking with respect to what?) or what is outperforming previous data.

---

> ### Author Response · Authors · 2024-12-30
> **Response to Reviewer GSa5**
>
> Thank you for the thoughtful review.
>
> - **Architecture differences:** We have added a figure (Fig. 5) to the appendix clarifying the differences between mean pooling architectures, models applying cosine or Euclidean distance on top of separate embeddings, and our Transformer aggregation architecture to support the verbal explanations in Section 3.
>
> - **Limited evaluation environments:** Meta-World is a standard and widely used benchmark for manipulation tasks, especially for task generalization, which makes our results easily interpretable and replicable by the robot learning community. Moreover, we also report reward learning results on VLMbench, a second robotic benchmark. Prior works have also chosen to evaluate on Meta-World and focused on simulated tasks (Sontakke et al., 2023). We have added a Limitations and future work section and have mentioned this there.
>
> - **Language descriptions:** The language descriptions we use are from the original Meta-World paper (the first sentences from Table 2 in Appendix A - Task Descriptions in Yu et al.). These are fairly succinct and intuitive (examples: “Rotate the faucet counter-clockwise”, “Press a button from the top”, “Pull a handle up”) and we did not need to modify them further. We have mentioned this choice of task descriptions in the updated PDF.
>
> - **Computational cost:** Using a reward model does indeed introduce some computational cost. However, as we use an off-policy reinforcement learning algorithm whose training data is sampled from a replay buffer, a requirement for real-time execution can be lifted and reward prediction can run entirely separately from the actor process, as is common in distributed RL setups. Running our reward model architecture (with only the current time step in the batch) takes up 800 MB of GPU RAM, and one forward pass (predicting the reward for one time step) takes 11 ms on a H100 GPU or 29 ms on a V100. We report these numbers in Appendix C.2 - Computational cost in the updated paper.
>
> - **Representative rollouts:** Thank you for the suggestion. We have included videos of representative policy rollouts over the course of training in the updated supplementary materials. In addition to the image observations, the videos display the task caption, and the corresponding rewards predicted by VLC. We further included example demonstration videos (successes and failures) in the updated supplementary materials. We will create a project webpage and also upload these videos there. We will also include image observations, task captions and reward predictions as images in the manuscript itself, along the lines of those in the supplementary videos.
>
> **Minor**:
> - We will fix the title alignment.
> - In related work, we actually discuss ranking trajectories in a different sense: here, we mean labeling trajectories with estimated rewards (even at the trajectory level). We have rephrased this to  “scoring the quality of existing trajectories”.
>
> **Requested changes**
>
> - We condition our model on 12 frames sampled at uniform intervals from the full video trajectory since the beginning of the episode. Conditioning on only the most recent history could be problematic for tasks defined relative to an initial position, such as “Move an object left” or “Rotate a dial 180 degrees”.
> - The MW50 experiment mainly serves as an evaluation of the ability of one reward model to represent well-formed reward functions for several language-conditioned tasks. It is not totally obvious a priori that this would work well. However, the focus of our experimental evaluation is on task generalization (MW40) and domain generalization (Open X). We agree that the VLC Open-X experiment is more representative of real-world applications.
> - Indeed, training on human data would be a relevant and promising direction for future work.
> - **Stability**: We did not observe instabilities in reward model training; both loss terms decrease over optimization steps when appropriately weighted.
>
>     **Consistency**: We have repeated the training of VLC MW40 with two new random seeds. Although there is variation across checkpoints for individual tasks, the averaged results over 6 tasks are very consistent, and the reported results are representative of the average of the replicated models. The results are included in the below table:
>
>     Model (MW40)|easy (2)|hard (4)|mean
>     -|-|-|-
>     VLC| 58 ± 22|79 ± 8|72 ± 11
>     VLC (2nd seed)|42 ± 24|85 ± 5|71 ± 13
>     VLC (3rd seed)|73 ± 5|73 ± 7|73 ± 5
>
> - Open-source code: The implementation is already included in the supplementary materials zip file to this submission. The pre-trained models along with the code will be open sourced.
> - Limitations and future work: Thank you for the suggestion, we have added a section on limitations and future work in the updated PDF.
>
> Please let us know if we have addressed your concerns or if any further clarifications would be useful.

---

> > ### Comment · Reviewer_GSa5 · 2024-12-30
> > **Thank you for your effort**
> >
> > I thank the authors for their efforts. All my concerns have been resolved and I think the modifcations and additions make the paper clearer and stronger. I acknowledge that I have read the other reviews and the authors' response. I'm happy to recommend accepting this paper.

---

### Review · Reviewer_TY2a · 2024-12-02

**Summary Of Contributions:**

This paper proposes a method called VLC which fine-tunes pre-trained vision language models to provide dense reward for training policies for robotics tasks. The training objective for VLC is the combination of a contrastive loss to increase the score of positive video-caption pairs and a sequential ranking loss to encourage later time steps to have higher scores. The main innovations in this work are the sequential ranking loss and using videos/sequences of frames rather than single frames as observations. The experiments demonstrate consistent performance gain over the sparse reward only setting.

**Audience:**

Yes

**Broader Impact Concerns:**

There's no broader impact concerns.

**Claims And Evidence:**

Yes

**Requested Changes:**

I have no requested changes but would like the authors to address the questions above.

**Strengths And Weaknesses:**

**Strengths**
* Presentation of the main ideas are clear and compact.
* The experiments are generally well organized and well executed.

**Weakness**
* It's not quite clear to me why a sparse reward is still added even when the VLC reward is used. If VLC were to be used as a general purpose reward, then it should be able to correctly identify when the observation sequence matches the task caption and this should be the only signal used to train a policy right? The weighted sum of sparse and VLC reward makes the method look more like reward shaping.
* I think the sequential ranking loss is a good idea for some tasks (e.g., goal reaching) but I don't think increasing rewards along a trajectory holds in general. A simple counter example is stochastic environments where environment noise could set the agent to a state less rewarding or optimal than before.
* How important are using multiple frames and using the sequential ranking loss? The architecturally closest and also the strongest baseline is Lorel (achieves almost the same scores on the easy tasks), which misses both multiple frames and sequential ranking loss. So it's hard to know which of the two innovations is the main contributor (or both) to the success of VLC. For multiple frames, it could be used by most if not all baseline methods in principle.
* When the authors say "broad coverage of execution history", is that referring to using both successful and failure demonstrations as training data? I thought all baseline methods are training on MW40 which contains failure data? Generally how important it is to incorporate the failure data for training?
* Could the authors provide more detail on the negative sample, .e.g, what exactly are used as negatives?

---

> ### Author Response · Authors · 2024-12-30
> **Response to Reviewer TY2a**
>
> Thank you for the thoughtful review.
>
> - **Requirement of sparse reward**: Entirely replacing the reward signal for a new task zero-shot using only language is still a very difficult problem, which all reward models we tested struggled with (no model exceeded 20% in 1-2M steps). A key challenge is the ambiguity of language: e.g., door-open expects a very large angle opening angle, whereas a text-conditioned model has no reason to think a small angle should not be considered a success. Future work could certainly explore making learned rewards even more performant, such as by improving reward models at policy training time, or learning explicit classifiers specializing in success detection without also having to capture task progress, as VLC does. Moreover, providing a sparse success reward, even if user-defined and not learned, is considerably easier and less error-prone than crafting a well-shaped dense reward.
>
> - **Goal-reaching rewards**: We argue many tasks can be posed as goal reaching. In the tasks we consider, the effect of stochasticity is rather small. Moreover, our methodology still allows for learning at the policy training stage. If there was, for example, a probability of transitioning to a worse state somewhere along what was initially deemed a promising trajectory, as long as the reward model recognizes the new state as a worse outcome, the policy’s value function can adjust to this drop and learn to avoid the stochastic state (since the learned model is used to predict rewards, not state values). We have added a mention of this in the Limitations section.
>
> - **Multiple frames and the sequential ranking loss**: Both using multiple frames and the sequential ranking loss are important, as demonstrated respectively by VLC’s improved performance over the LOReL baseline (which uses 2 frames only) and over both RoboCLIP and the VLC alpha = 0 ablation (which use only the contrastive objective without the sequence ranking loss).
>
>     We would like to emphasize (as mentioned in Appendix E and Table 6) that in our experiments, we reimplemented the LOReL baseline using our architecture and CLIP pretraining for a fairer comparison, as the original LOReL used a considerably weaker architecture and no visual pretraining. To clarify this, we now highlight this implementation difference to the original LOReL already in Section 4.4. As a result, the expressiveness of our architecture also contributed to this baseline performing as strongly as it did. Despite benefiting for our more performant architecture, results for LOReL in 4/6 tasks (open door, slide plate, press button, pull handle) were all significantly below the confidence interval for VLC. For a thorough comparison and high statistical significance, the experiments in Fig. 3 / Table 2 were run with 10 random seeds for VLC, LOReL and RoboCLIP. LOReL indeed performs on par with VLC on the easy tasks, but please note that there are only 2 tasks in this set, whereas on the 4 hard tasks and on average across all 6 tasks, VLC outperforms LOReL with a significant margin: **79 vs. 60** for the hard tasks and **72 vs. 59** on all tasks.
>
>     **Ablation on the sequence ranking loss**: To quantify the contribution of the sequence ranking loss, we have added an ablation study without it. The results are included in the revised PDF (Appendix B), and summarized in the below table:
>
>     Method (MW40)|easy (2)|hard (4)|mean
>     -|-|-|-
>     VLC | 58 ± 22|79 ± 8|**72** ± 11
>     VLC (no seq. ranking)|67 ± 15|66 ± 7|66 ± 7
>
>     As there are only two tasks in the easy task set, results on the intermediate set and the average across all tasks are more meaningful. The gains over _VLC (no seq. ranking)_ are +13 on the intermediate set and +6 percentage points across all tasks, which is small but consistent with the gains we observed on VLMBench data, our second evaluation benchmark (see Appendix A). Furthermore, it is possible that tuning the sequence ranking weight on Meta-World would further improve performance, but we did not explore this and instead tuned all hyperparameters on VLMbench only.
>
> - **”Broad coverage of execution history”**: By this claim, we mean that our reward model is conditioned on 12 frames, whereas all prior methods except RoboCLIP considered only 1 or 2. For clarity, we have rephrased this to the following: “Thanks to its conditioning on more frames of execution history and the sequential loss term --”
>
> - **Negative samples**: We generate failure videos by mixing a demonstrator’s actions with noise (with probability 0.7, the demonstrator’s action is replaced with a random action as described in Section 4.1). We have included examples of the failure (and success) videos in the updated supplementary materials. However, other approaches to generating failures could also be leveraged and our approach is in no way specific to this choice.
>
> Please let us know if we have addressed your concerns or if any further clarifications would be useful.

---

> > ### Comment · Reviewer_TY2a · 2025-01-01
> >
> > Thank the authors for the clarifications and additional experiments. All my questions have been addressed and the paper is much clearer. I recommend accepting this paper.

---

### Review · Reviewer_jBHS · 2024-12-09

**Summary Of Contributions:**

The paper introduces Video-Language Critic (VLC) a new approach to learn a goal conditioned similarity functions where the observation is a video and the goal is textual.
Compared to LOREL or VIP/LIV, the main novelty is the new architecture that takes as input multiple frames and the new loss.
This new loss is composed of a cross entropy loss with a new sequential ranking loss that assume that the reward should always increase in the observed trajectory.

**Audience:**

Yes

**Claims And Evidence:**

No

**Requested Changes:**

- I think an ablation study on the proposed loss L_seq is necessary.

- It would also be interesting to train your new architecture with LIV losses on the same training dataset to be sure the proposed loss is more suitable for video inputs.

- For LOREL, why Num. images in Table 5 isn't also 12? Does "Num. Images" is the value of N in (2)?

**Strengths And Weaknesses:**

## Strengths:
- the approach is more straightforward than LIV
- it is natural to consider multiple frames as input in this setting

## Weaknesses:
- in the "sequential ranking objective" paragraph, the claim that prior reward learning approaches ignore the relative ranking of states is not supported by evidence. In LIV/VIP, the value of the next state also appears in the loss as in your L_seq loss
- the assumption that the reward should always increase means that the observed trajectories are always optimal, however, it's often the case that the demonstration contains suboptimal subparts
- the experiments are not very convincing because:
    *  it is not clear which of the main two contributions is more important, I am not sure if the proposed loss is essential or is it the new architecture
    * different models are not trained on the same dataset (for instance LIV), therefore it is hard to properly do a comparison

---

> ### Author Response · Authors · 2024-12-30
> **Response to Reviewer jBHS**
>
> Thank you for the thoughtful review.
>
> - **Relative ranking of states**: As you noted, LIV and VIP do leverage the _ordering_ of intermediate states in that they source negative pairs for the contrastive objective from temporally neighbouring states. However, their objective leverages no assumptions about the relative _ranking_ of these states, as ours does (their objective does not have any prior on the later state having a higher value, and thus does not enforce any relative ranking or monotonous improvement). Therefore, we think the statement in question is justified.
>
>     We will make this claim even more explicit:
>     “Video inputs also contain implicit information about the relative ranking of states, which is not leveraged in prior reward learning approaches (Fan et al., 2022; Nair et al., 2022a;b; Karamcheti et al., 2023; Ma et al., 2023a; Sontakke et al., 2023). While Ma et al., 2023a, use order information, they do not use relative ranking: consecutive frames are used bidirectionally as negatives in their contrastive objective, with their representations pushed apart from each other. This encourages the frames’ representations to vary gradually over the video time span but does not leverage any prior on the later state having a higher value.”
>
> - **Reward increasing for observed trajectories**: It is true that the sequence ranking objective most benefits from optimal or high-quality trajectories. However, this is also true for imitation learning approaches, and as mentioned towards the end of the introduction, learning a reward function (unlike imitation learning) still allows the policy to outperform the existing trajectories by executing the task faster or by finding better solutions: the highest rewards are still assigned to the final states, so getting to one faster is encouraged. Moreover, our objective does not impose consistent increase, only non-decrease of rewards. Therefore, a suboptimal section of a demonstration can be assigned a constant reward, representing non-progress in the task, and the objective only assumes that the suboptimal parts do not make negative
> progress in the task, which is a much less restrictive assumption. We have added a Limitations section to the revised manuscript and mention this assumption there.
>
> - **Ablation experiments**: It is indeed valuable to
>   - separately evaluate the contributions of the sequence ranking loss and of the architecture, and to
>   - consider our architecture trained with the LIV loss.
>
>   We have run both ablation studies, as requested. The full VLC objective outperforms both ablations on the 6 Meta-World held-out tasks. All policy training runs were repeated with 10 random seeds. The results are included in the revised PDF, and summarized in the below table:
>
>     Method (MW40)|easy (2)|hard (4)|mean
>     -|-|-|-
>     VLC | 58 ± 22|79 ± 8|**72** ± 11
>     VLC (no seq. ranking)|67 ± 15|66 ± 7|66 ± 7
>     LIV loss, VLC architecture|46 ± 20|63 ± 11|58 ± 10
>
>     As there are only two tasks in the easy task set, results on the intermediate set and the average across all tasks are more meaningful. The gains over _VLC (no seq. ranking)_ are +13 on the intermediate set and +6 percentage points across all tasks, which is small but consistent with the gains we observed on VLMBench data, our second evaluation benchmark (see Appendix A). Furthermore, it is possible that tuning the sequence ranking weight on Meta-World would further improve performance, but we did not explore this and instead tuned all hyperparameters on VLMbench only.
>
>     Note that applying the LIV loss to our architecture requires the time contrastive loss on images to be applied after the visual encoder but before the temporal aggregator, whereas the vision-language contrastive loss is applied to the final output.
>
> - **Number of images for LOReL**: LOReL only considered conditioning on start and end frames; we maintained this choice for our baseline for consistency with the original work. Moreover, the value of N in Eq. (2) is not the number of frames but the batch size (64 in our experiments).
>
> Please let us know if we have addressed your concerns or if any further clarifications would be useful.

---

> > ### Comment · Reviewer_jBHS · 2025-01-07
> >
> > Thank you for answering my concerns and running the additional experiments.

---

### Author Response · Authors · 2024-12-30
**Author response to reviewers, updates to manuscript and supplementary materials**

We would like to thank all reviewers for their thoughtful and constructive feedback. We address the individual review comments in the respective responses below. We have also revised our manuscript accordingly and updated the PDF with the following key additions:
1. **Additional experiments & ablations** (Appendix B)
    - **Sequence ranking loss ablation**: As requested by reviewers jBHS and TY2a, we have added an ablation of our method without the sequence ranking loss, and
    - **LIV loss with VLC architecture**: we have trained our architecture with LIV’s time-contrastive loss.
    - **The full VLC objective outperforms both ablations**, confirming that both the proposed architecture and the sequence ranking loss contribute to improved performance.
2. New Limitations and future work section (Section 5)
3. Improved visualizations comparing our architecture to alternatives (Fig. 5)
4. Added details on inference-time computational cost (Appendix C)
5. Clarified the source of the task language descriptions (Appendix D)

When reviewing the updated manuscript, please note the highlighted changes in the Appendix in addition to those in the main body, especially the experiments in the new Appendix B.

We have also added **videos of reward predictions and policy rollouts** to the supplementary materials, and **repeated VLC reward model training twice** using two new random seeds **with results on par** with our reported experiments, as requested by reviewer GSa5.

We believe that the contributions of our work are further strengthened by the additional ablations, and that the presentation clarity of the paper has been improved with the new visualizations, limitations section, and training details.

---

### Decision · Action_Editor_J6oh · 2025-01-24

**Recommendation:** Accept as is

**Comment:**

The discussion clarified the differences with respect to previous work and the additional computation cost. The ablation of the contribution of the sequential ranking loss is an important addition.

After the discussion, there is a consensus among the reviewers to accept the paper. I support the acceptance as well.

**Audience:**

Learning robotics policies based on videos is relevant to a wide audience.

**Claims And Evidence:**

The paper introduces a language conditional model that, given a video segment and its caption, returns scores that can be used as a policy for a robot. A novel loss function includes a contrastive component for video and caption, but that doesn’t consider the order of the states, as it’s adding over the distance between pairs of states (no captions). The novel loss function includes a term for encouraging the completion of the task in order, so it considers the captions. As the function trained with the proposed objective can be used to obtain a reward, it can follow the trajectories of the data, but it could be adjusted to consider that some trajectories are suboptimal. This confirms the first two claims in the list of contributions. The experimental results confirmed the other three claims.

The reviewers value the problem and the clarity of the presentation, including the experiments and their discussion. Some reviewers praised the improved sample efficiency and the generalization capabilities. The experiments on MetaWorld are limited but significant. As the loss function has separated components

The reviewers have doubts about the formulation of the sequential ranking objective. There were also questions about the relationship of the contribution with previous work. Some reviewers were skeptical about the assumption that rewards are increasing. However, I’m convinced of the value of narrowing the scope to goal-oriented tasks. Moreover, it doesn’t matter if the sequences in the data are suboptimal, as the objective is to match the data.